# Waist circumference and mortality or cardiovascular events in a general Korean population

**Do Kyeong Song, Young Sun Hong, Yeon-Ah Sung, Hyejin Lee** *

Department of Internal Medicine, Ewha Womans University School of Medicine, Seoul, Korea

* hyejinlee@ewha.ac.kr

**Data Availability Statement:** All relevant data are within the manuscript.

**Funding:** The authors received no specific funding for this work.

## Abstract

### Background

Obesity is associated with cardiovascular diseases and is a risk factor for all-cause mortality. Until now, the associations between abdominal obesity and mortality or cardiovascular disease (CVD) incidence have not been conclusive. We aimed to evaluate the associations between waist circumference (WC) and mortality or CVD incidence in a general Korean population.

### Methods

We analyzed a total of 204,068 adults older than 40 years of age who had undergone a national health examination at least once from 2009 to 2018 in the Korean National Health Insurance Service Cohort. WC was divided into five categories (< 80, 80–84.9, 85–89.9, 90–94.9, $\geq$ 95 cm). Hazard ratios for death and CVD incidence were calculated using Cox proportional hazards models.

### Results

In men, WC and overall mortality showed a reverse J-shaped association. In women, the association between WC and overall mortality was not significant. For both men and women, WC was not associated with the risk of cardiovascular mortality. Contrary to the mortality trend, CVD incidence was positively associated with WC in both men and women, and the risk of the CVD incidence was the lowest in subjects with a WC < 80 cm.

### Conclusions

WC exhibited a significant J-shaped association with overall mortality in men, where subjects who had central obesity showed a lower rate of mortality than those in the lowest or highest WC group. The risk of incident CVD showed a positive association with central obesity, where the lowest risk was observed for subjects in the lowest WC group in a general Korean population.

**Competing interests:** The authors have declared that no competing interests exist.

## Introduction

The prevalence of overweight or obesity has increased, and obesity is an important risk factor for cardiovascular diseases, type 2 diabetes mellitus and several cancers [1]. Therefore, excess body weight was suggested to be associated with health problems, including premature death [2–4]. However, the relationship between obesity and mortality is inconclusive. Recently, data from a large cohort have suggested the concept of the obesity paradox, in which overweight or obese subjects exhibited a better prognosis than leaner subjects [5, 6]. Additionally, several studies have evaluated the relationship between obesity and mortality in Korea. A 12-year prospective cohort study of 1,213,829 Koreans between the ages of 30 and 95 years in the Korean Cancer Prevention Study showed that body mass index (BMI) exhibited a J-shaped association with all-cause mortality [7]. Among 415,796 Korean adults older than 30 years who had undergone a national health examination, BMI showed a U-shaped association with overall mortality [8]. Although BMI is the most commonly used anthropometric measure to assess obesity, an important limitation of BMI is its inability to assess the distribution of body fat. Additionally, in older adults, BMI may not be useful to measure adiposity because fat tends to be redistributed with aging toward an increase in abdominal fat [9].

Waist circumference (WC) is a measure of abdominal obesity and is known to be strongly correlated with visceral adipose tissue. Visceral fat is commonly used to describe intra-abdominal fat and includes both intraperitoneal fat (mesenteric and omental fat) and retroperitoneal fat [10]. Recent studies have suggested that increased visceral fat or WC is more predictive of all-cause mortality than the actual total amount of body fat or BMI [11, 12]. In particular, visceral adipose tissue is known to be strongly associated with an adverse metabolic risk profile [13]. High levels of WC have been reported to be associated with metabolic diseases such as type 2 diabetes mellitus, dyslipidemia, and coronary heart disease [10].

To date, there have been several studies about the association between WC and mortality; however, the results have been inconsistent. Increased WC was associated with a higher risk of mortality after adjustment for BMI in a large US cohort including subjects with comorbidities such as hypertension, diabetes, cardiovascular disease, cancer, and respiratory disease [14]. A J-shaped association of WC with all-cause and cardiovascular mortality was found in a meta-analysis of 29 cohorts involving predominantly Caucasian populations after excluding those with major chronic diseases such as cardiovascular disease (CVD), cancer, and respiratory diseases at baseline [15]. WC showed a positive linear association with all-cause mortality after adjusting for BMI in the Korean population of individuals older than 20 years who underwent the National Health Insurance Service (NHIS) health checkup from 2009 to 2015 (23,263,878 subjects) [16]. However, chronic diseases such as diabetes mellitus, hypertension, and dyslipidemia were not adjusted in the study.

Given the unclear association between WC and overall mortality, more research in a large population with a longer follow-up period is needed. Furthermore, there have been few studies about the association between WC and CVD mortality or CVD incidence. We aimed to evaluate the associations between WC and mortality or CVD incidence in a general Korean population using a cohort database based on Korean NHIS data between 2009 and 2018.

## Materials and methods

### Data source

We used the cohort database released by the NHIS. The NHIS cohort can represent the Korean general population because the Korean National Health Insurance Program is a universal

health insurance program in Korea; the Korean National Health Insurance Program was initiated in 1963 and is mentioned in detail elsewhere [8]. Approximately 97% of the Korean population is enrolled in the NHIS. All individuals older than 40 years are invited to participate in a biannual health checkup as part of the NHIS. The participants completed a self-administered questionnaire that included information on demographic, medical, and behavioral factors.

We did not obtain informed consent from the participants because the data were not collected for the study. The patient records from the NHIS were anonymous before being released by the NHIS. This study was approved by the Institutional Review Board (IRB) of the Ewha Medical Center.

## Study population

We analyzed subjects older than 40 years of age who had undergone a national health examination at least once from 2009 to 2018 in the Korean NHIS cohort database (n = 449,605). The follow-up period started from the first health examination. Subjects who had missing data for total cholesterol (n = 144), current smoking status (n = 7,075), alcohol consumption status (n = 4,020), physical activity level (n = 5,784), or family history (n = 154,198) were excluded. Subjects who had a past medical history of stroke (n = 36,390), ischemic heart disease (n = 48,236), cancer (n = 60,669), or chronic obstructive pulmonary disease (n = 18,161) were excluded. Subjects who died within 1 year (n = 2,026) were excluded. Finally, 204,068 subjects (men = 112,449 and women = 91,619) were included in this study.

## Outcome variables and covariates

WC was measured in a standing position at the point midway between the lower costal margin and the iliac crest. WC was divided into five categories ($<$ 80, 80–84.9, 85–89.9, 90–94.9, $\geq$ 95 cm). The reference WC was $<$ 80 cm. Age was categorized into 10-year groups (40–49, 50–59, 60–69, and $\geq$ 70 years). Current smoking status was classified into smoking or non-/ex-smoker. Alcohol consumption status was classified into heavy drinking ($>$ 2 drinks/day in men and $>$ 1 drink/day in women) or nonheavy drinking. Physical activity level was grouped into 3 groups: none, 1–2 times/week, and $\geq$ 3 times/week.

Combined conditions such as hypertension and diabetes mellitus were identified based on questionnaire responses or medical claim data. Total cholesterol was classified into three groups ($<$ 200, 200–239, and $\geq$ 240 mg/dl).

The outcomes were time to death, time to CVD death, and time to CVD incidence. CVD incidence was defined as the first admission with a diagnosis of CVD. CVD was identified based on ICD-10 codes (ICD codes I20-25 for ischemic heart disease, ICD codes I60-62 for hemorrhagic stroke, and ICD codes I63 for ischemic stroke).

## Statistical analyses

Categorical variables are presented as frequencies and proportions. We examined the associations between categorical WC and all-cause and CVD mortality using Cox proportional hazards models after adjustment for age, smoking, alcohol consumption status, levels of physical activity, total cholesterol, hypertension, and diabetes mellitus status. Only age was adjusted in model 1; age, smoking status, alcohol consumption status, and level of physical activity were adjusted in model 2; and age, smoking status, alcohol consumption status, level of physical activity, total cholesterol, hypertension, and diabetes mellitus status were further adjusted in model 3.

The analyses were conducted separately for men and women. *P* values < 0.05 were considered statistically significant. All statistical analyses were performed using SAS (version 9.4, SAS Institute, Cary, NC).

## Results

We analyzed the data of 204,068 subjects (men = 112,449 and women = 91,619) in the study. Table 1 shows the demographic and clinical characteristics of the subjects according to WC category. The proportions of subjects in the WC groups (< 80, 80–84.9, 85–89.9, 90–94.9, ≥ 95 cm) were 38.1%, 24.9%, 19.9%, 11.1%, and 6.0%, respectively. As WC increased, age and the proportion of men, current smokers, heavy drinkers, subjects with high levels of cholesterol, and subjects with hypertension or diabetes mellitus increased.

During 10 years of follow up, 3,528 deaths occurred. Among them, 412 were because of CVD. During the follow-up period, CVD occurred in 11,687 subjects. As WC increased, the proportion of subjects who died or had CVD increased.

Table 2 shows the association between WC and overall mortality by sex. For men, WC and overall mortality showed a reverse J-shaped association, with the highest mortality rate among those with a WC below 80 cm in models 1, 2, and 3 (Fig 1A). The overall mortality rate was the lowest among those with a WC of 90–94.9 cm after adjustment for age, smoking, alcohol consumption status, level of physical activity, total cholesterol, hypertension, and diabetes mellitus status in men. Women with the highest WC showed the highest overall mortality in models 1 and 2. However, the association between WC and overall mortality in women was not

**Table 1. Demographic and clinical characteristics of the subjects according to waist circumference categories.**

| | Waist circumference groups | | | | |
|---|---|---|---|---|---|
| | < 80 (n = 77,866) | 80–84.9 (n = 50,725) | 85–89.9 (n = 40,555) | 90–94.9 (n = 22,746) | ≥ 95 (n = 12,176) |
| **Age (years)** | | | | | |
| - 40–49 | 12,740 (16.3%) | 7,279 (14.3%) | 5,509 (13.6%) | 2,799 (12.3%) | 1,343 (11.1%) |
| - 50–59 | 41,167 (52.9%) | 25,396 (50.1%) | 20,151 (49.7%) | 10,930 (48.1%) | 5,638 (46.3%) |
| - 60–69 | 16,331 (21.0%) | 12,647 (24.9%) | 10,372 (25.6%) | 6,104 (26.8%) | 3,401 (27.9%) |
| -≥70 | 7,628 (9.8%) | 5,403 (10.7%) | 4,523 (11.1%) | 2.913 (12.8%) | 1,794 (14.7%) |
| Male | 26,294 (33.8%) | 30,897 (60.9%) | 29,097 (71.8%) | 17,035 (74.9%) | 9,126 (75.0%) |
| Smoking status | | | | | |
| -current smoker | 10,570 (13.6%) | 10,370 (20.4%) | 9,334 (23.0%) | 5,360 (23.6%) | 2,827 (23.2%) |
| -non-/ex-smoker | 67,296 (86.4%) | 40,355 (79.6%) | 31,221 (77.0%) | 17,386 (76.4%) | 9,349 (76.8%) |
| Alcohol consumption status | | | | | |
| -Heavy drinker | 9,119 (11.7%) | 8,730 (17.2%) | 8,152 (20.1%) | 4,931 (21.7%) | 2,831 (23.3%) |
| -Nonheavy drinker | 68,747 (88.3%) | 41,995 (82.8%) | 32,403 (79.9%) | 17,815 (78.3%) | 9.345 (76.8%) |
| Physical activity | | | | | |
| -None | 37,820 (48.6%) | 22,607 (44.6%) | 17,662 (43.6%) | 10,246 (45.1%) | 5,805 (47.7%) |
| -1-2 times/week | 18,691 (24.0%) | 13,532 (26.7%) | 11,420 (28.2%) | 6,285 (27.6%) | 3,215 (26.4%) |
| - ≥ 3 times/week | 21,355 (27.4%) | 14,586 (28.8%) | 11,473 (28.3%) | 6,215 (27.3%) | 3,156 (25.9%) |
| Total cholesterol < 200 mg/dl | 41,009 (52.7%) | 25,152 (49.6%) | 19,909 (49.1%) | 11,003 (48.4%) | 5,926 (48.7%) |
| Total cholesterol 200–239 mg/dl | 26,724 (34.3%) | 18,082 (35.6%) | 14,488 (35.7%) | 8,158 (35.9%) | 4,347 (35.7%) |
| Total cholesterol ≥ 240 mg/dl | 10,133 (13.0%) | 7,491 (14.8%) | 6,158 (15.2%) | 3,585 (15.7%) | 1,903 (15.6%) |
| Hypertension | 14,926 (19.2%) | 14,082 (27.8%) | 13,604 (33.5%) | 9,017 (39.6%) | 5,937 (48.8%) |
| Diabetes mellitus | 8,439 (10.8%) | 7,672 (15.1%) | 7,099 (17.5%) | 4,630 (20.4%) | 3,226 (26.5%) |

Values are presented as the frequencies and proportions.

**Table 2. Hazard ratio for overall mortality according to waist circumference categories.**

| | | Waist circumference groups | | | | |
|---|---|---|---|---|---|---|
| | | < 80 | 80–84.9 | 85–89.9 | 90–94.9 | ≥ 95 |
| Men | Model 1 | 1.000 | 0.720* | 0.743* | 0.726* | 0.777* |
| | Model 2 | 1.000 | 0.754* | 0.784* | 0.771* | 0.829* |
| | Model 3 | 1.000 | 0.740* | 0.759* | 0.733* | 0.773* |
| Women | Model 1 | 1.000 | 0.901 | 0.949 | 1.045 | 1.447* |
| | Model 2 | 1.000 | 0.899 | 0.946 | 1.032 | 1.416* |
| | Model 3 | 1.000 | 0.857 | 0.877 | 0.932 | 1.223 |

* $P < 0.05$ compared to subjects with waist circumference below 80 .

Model 1: adjusted for age. Model 2: adjusted for age, smoking status, alcohol consumption status, and level of physical activity. Model 3: adjusted for age, smoking status, alcohol consumption status, level of physical activity, total cholesterol, hypertension, and diabetes mellitus status.

significant after adjustment for age, smoking, alcohol consumption status, level of physical activity, total cholesterol, hypertension, and diabetes mellitus status (Table 2, Fig 1B). For both men and women, WC was not associated with the risk of cardiovascular mortality after adjustment for age, smoking, alcohol consumption status, level of physical activity, total cholesterol, hypertension, and diabetes mellitus status. Women with the highest WC showed the highest CVD mortality only in models 1 and 2 (Table 3, Fig 1C and 1D).

We analyzed CVD incidence according to WC categories (Fig 2). Contrary to the mortality trend, the CVD incidence was positively associated with WC in both men and women, and the risk of the CVD was the lowest in subjects with a WC < 80 cm in models 1, 2, and 3. The association between WC and ischemic heart disease incidence was similar to the association between WC and CVD incidence. For ischemic stroke incidence, men with a WC of 90–94.9 cm showed a higher risk than men with a WC below 80 cm. In women, there was a trend showing a positive association between WC and ischemic stroke incidence, although the association was not statistically significant in model 3. For hemorrhagic stroke, WC was negatively associated with stroke incidence for men, and WC was not associated with the stroke incidence for women (Table 4).

## Discussion

During the 10 years of follow-up, 3,528 deaths occurred. WC was significantly associated with overall mortality in men with a reverse J-shaped association but not significantly associated in women. The risk of CVD incidence showed a positive association with central obesity for both men and women, where the lowest risk was observed for subjects in the lowest WC group in a general Korean population.

In our study, there was a reverse J-shaped association between WC and overall mortality for men. Consistent with the results of our study, WC showed a J-shaped or U-shaped association with mortality after adjustment for comorbidities among 8,796,759 Korean subjects aged between 30 and 90 years; in normal-weight and overweight women, the relationship was J-shaped, whereas in overweight men and obese subjects, the relationship was U-shaped [17]. Additionally, among elderly persons aged 65–74 years in a predominantly Caucasian population, WC showed a J-shaped association with all-cause mortality after excluding those with major chronic diseases such as cardiovascular disease, cancer, and respiratory disease [15]. Among 154,776 men and 90,757 women aged 51–72 years who resided in US states, the association between WC and mortality was J-shaped [18]. In a systematic review and meta-

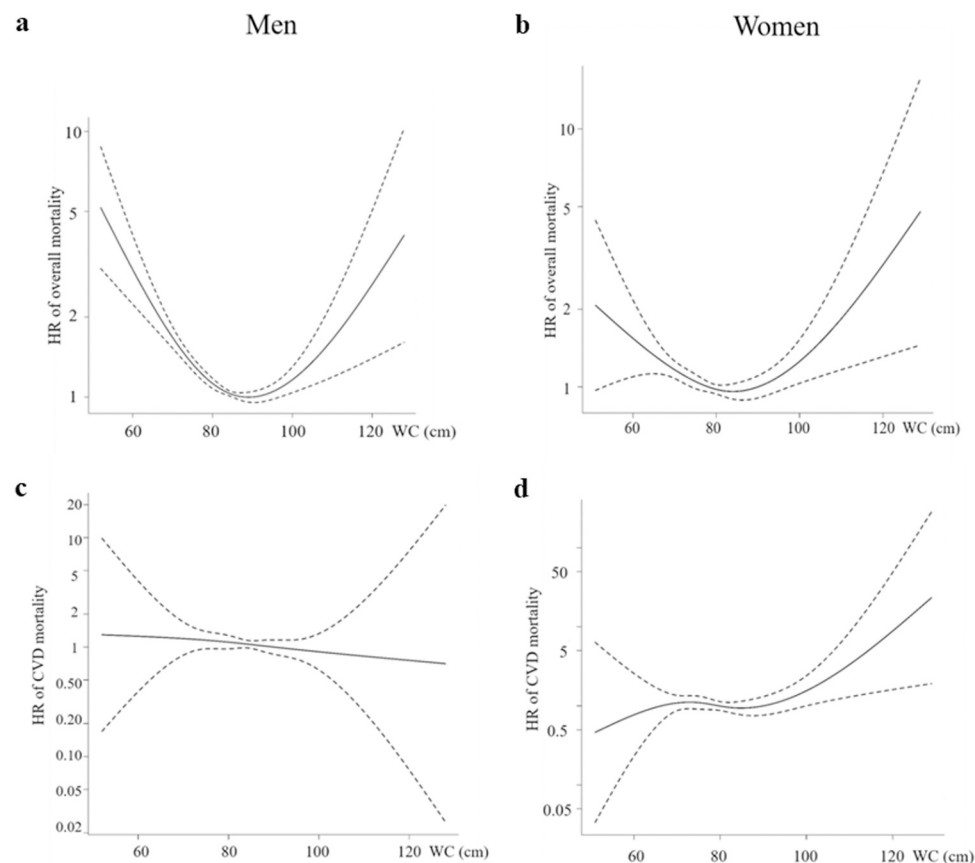

**Fig 1. Association between waist circumference as a continuous variable and overall mortality and CVD mortality as a cubic spline after adjusting for age.** For overall mortality and CVD mortality, the reference WC was set as <80 cm. CVD, cardiovascular disease; HR, hazard ratio; WC, waist circumference.

regression analysis comprising 689,465 participants during 5–24 years of follow-up, WC showed U- or J-shaped associations with mortality [19].

However, there were several studies demonstrating that WC was positively associated with mortality, which is not in agreement with the results of our study. There was a linear

**Table 3. Hazard ratio for cardiovascular disease mortality according to waist circumference categories.**

| | | Waist circumference groups | | | | |
|---|---|---|---|---|---|---|
| | | < 80 | 80–84.9 | 85–89.9 | 90–94.9 | ≥ 95 |
| Men | Model 1 | 1.000 | 0.935 | 1.059 | 0.946 | 0.834 |
| | Model 2 | 1.000 | 0.983 | 1.124 | 1.013 | 0.897 |
| | Model 3 | 1.000 | 0.890 | 0.971 | 0.841 | 0.710 |
| Women | Model 1 | 1.000 | 0.841 | 1.121 | 1.147 | 2.074* |
| | Model 2 | 1.000 | 0.838 | 1.116 | 1.125 | 2.008* |
| | Model 3 | 1.000 | 0.770 | 0.980 | 0.943 | 1.602 |

* $P < 0.05$ compared to subjects with waist circumference below 80 .

Model 1: adjusted for age. Model 2: adjusted for age, smoking status, alcohol consumption status, and level of physical activity. Model 3: adjusted for age, smoking status, alcohol consumption status, level of physical activity, total cholesterol, hypertension, and diabetes mellitus status.

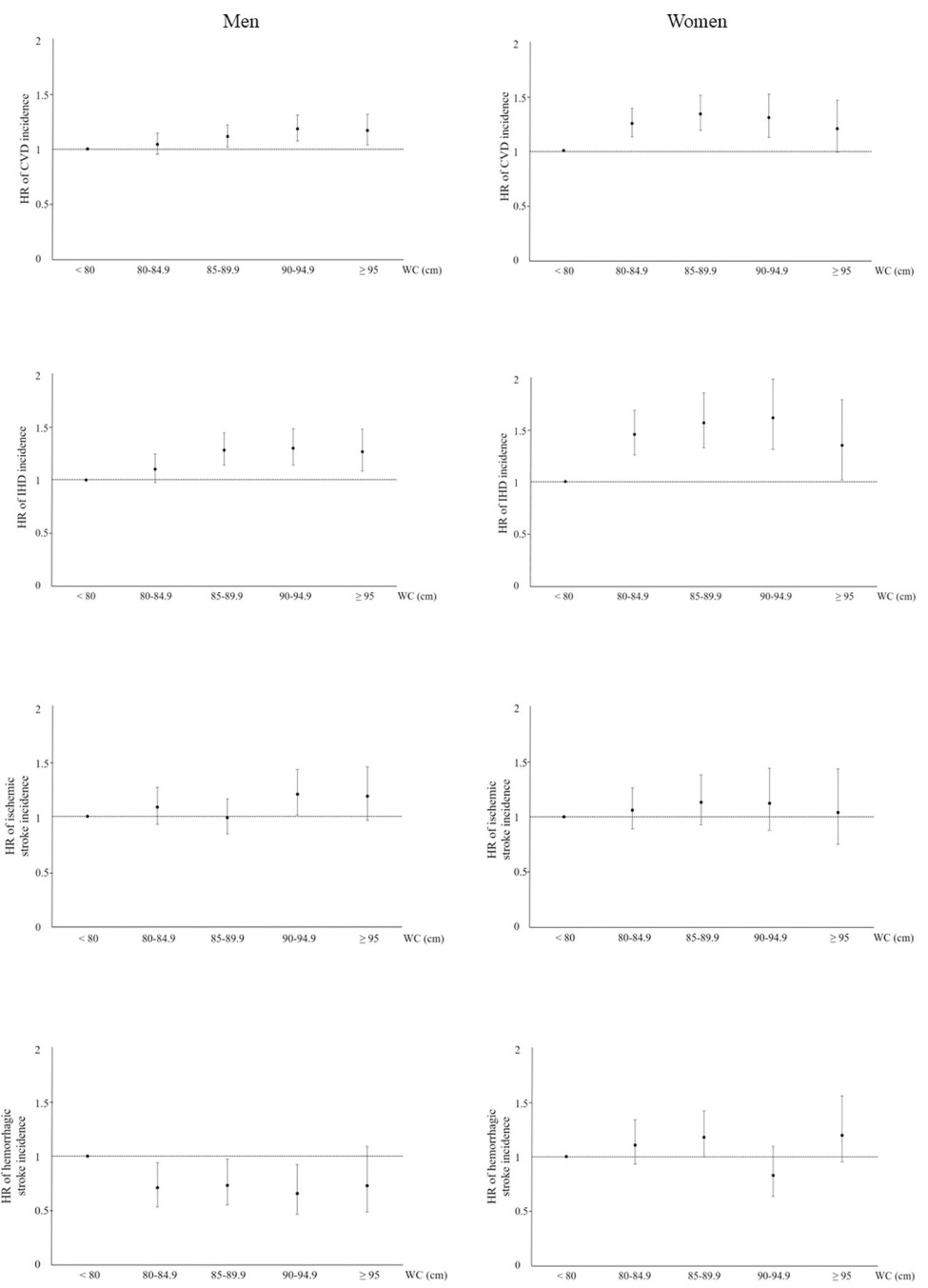

**Fig 2. Cardiovascular disease incidence according to waist circumference categories.** HRs were adjusted for age, smoking status, alcohol consumption status, level of physical activity, total cholesterol, hypertension, and diabetes mellitus status. WC, waist circumference; CVD, cardiovascular disease; HR, hazard ratio.

association between WC and all-cause mortality among a Korean population of individuals older than 20 years [16]. WC was positively associated with mortality in a large US cohort aged 50 years or older [14]. Neither of the studies adjusted for chronic diseases such as diabetes mellitus and hypertension as we did in our study. Additionally, there were several studies demonstrating a positive association between WC and mortality in middle-aged and elderly

**Table 4. Hazard ratio for cardiovascular events according to waist circumference categories.**

| | | | Waist circumference groups | | | | |
|---|---|---|---|---|---|---|---|
| | | | < 80 | 80–84.9 | 85–89.9 | 90–94.9 | ≥ 95 |
| Cardiovascular events | Men | Model 1 | 1.000 | 1.089 | 1.203* | 1.324* | 1.369* |
| | | Model 2 | 1.000 | 1.124* | 1.250* | 1.376* | 1.427* |
| | | Model 3 | 1.000 | 1.044 | 1.116* | 1.184* | 1.167* |
| | Women | Model 1 | 1.000 | 1.340* | 1.494* | 1.520* | 1.485* |
| | | Model 2 | 1.000 | 1.337* | 1.487* | 1.507* | 1.467* |
| | | Model 3 | 1.000 | 1.249* | 1.337* | 1.304* | 1.200 |
| Ischemic heart disease | Men | Model 1 | 1.000 | 1.172* | 1.419* | 1.496* | 1.536* |
| | | Model 2 | 1.000 | 1.200* | 1.460* | 1.543* | 1.591* |
| | | Model 3 | 1.000 | 1.102 | 1.282* | 1.300* | 1.266* |
| | Women | Model 1 | 1.000 | 1.588* | 1.799* | 1.945* | 1.748* |
| | | Model 2 | 1.000 | 1.587* | 1.795* | 1.938* | 1.735* |
| | | Model 3 | 1.000 | 1.453* | 1.565* | 1.612* | 1.347* |
| Ischemic stroke | Men | Model 1 | 1.000 | 1.091 | 1.014 | 1.272* | 1.298* |
| | | Model 2 | 1.000 | 1.148 | 1.079 | 1.356* | 1.389* |
| | | Model 3 | 1.000 | 1.082 | 0.986 | 1.200* | 1.181 |
| | Women | Model 1 | 1.000 | 1.129 | 1.251* | 1.296* | 1.273 |
| | | Model 2 | 1.000 | 1.126 | 1.243* | 1.280* | 1.250 |
| | | Model 3 | 1.000 | 1.058 | 1.130 | 1.123 | 1.037 |
| Hemorrhagic stroke | Men | Model 1 | 1.000 | 0.731* | 0.778 | 0.720 | 0.839 |
| | | Model 2 | 1.000 | 0.746* | 0.795 | 0.733 | 0.850 |
| | | Model 3 | 1.000 | 0.708* | 0.731* | 0.655* | 0.727 |
| | Women | Model 1 | 1.000 | 1.128 | 1.213 | 0.863 | 1.257 |
| | | Model 2 | 1.000 | 1.121 | 1.204 | 0.853 | 1.240 |
| | | Model 3 | 1.000 | 1.107 | 1.178 | 0.827 | 1.196 |

* $P < 0.05$ compared to subjects with waist circumference below 80 .

Model 1: adjusted for age. Model 2: adjusted for age, smoking status, alcohol consumption status, and level of physical activity. Model 3: adjusted for age, smoking status, alcohol consumption status, level of physical activity, total cholesterol, hypertension, and diabetes mellitus status.

individuals who were older than the subjects in our study. WC showed a strong dose-response-type relationship with mortality in men and women 50 to 64 years of age who were recruited in the Danish prospective study that adjusted for BMI [20]. In a previous study including Chinese individuals older than 50 years without medical conditions, greater WC was associated with an increased risk of all-cause mortality [21]. The inclusion criteria or age of the participants may affect the difference in the results between studies regarding the association between WC and overall mortality.

There were a few studies with a smaller sample size than our study showing an inverse association between WC and mortality. Among 3,554 men and 4,472 women (aged between 40 and 90 years) who had no history of ischemic heart disease or stroke in the general Japanese population over a follow-up period of 14.7 years, WC was inversely associated with all-cause mortality in men but not in women [22]. A 22-year cohort study including 15,582 participants aged 18 years or older from the China Health and Nutrition Survey found that lower WC was associated with a higher risk of all-cause mortality [23]. Among a total of 4,361 Chinese oldest old individuals (aged 80 years or older), WC was linearly associated with lower mortality in men and women over a 3-year period [24]. Because of the small sample sizes in the previous

studies, it is difficult to generalize the result of the inverse association between WC and mortality.

Because visceral fat is known to be a strong predictor of dyslipidemia and insulin resistance [25], it is possible that WC is associated with premature death resulting from CVD. In the Canadian Heart Health Follow-up Study, WC positively predicted all-cause, CVD, and cancer mortality over a mean 13-year follow-up among 8,061 adults (aged 18–74 years) [26]. Among 225,712 US women and men aged 50 to 71 years, higher WC was related to a higher risk of death from CVD, including coronary heart diseases and strokes [27]. Among 24,508 European men and women 45 to 79 years of age, during a mean 9.1 years of follow-up, HRs for coronary heart disease increased with WC [28]. In previous studies in Western populations, WC was linearly associated with CVD incidence; however, WC was not associated with CVD mortality for either men or women in our study. In our study, the associations between WC and ischemic heart disease incidence were similar to the association between WC and overall CVD incidence. Therefore, ischemic heart disease may be attributed to the linear association between WC and overall CVD incidence. The different results regarding the association between WC and mortality might be caused by ethnic differences. Asians have a larger amount of visceral fat than Caucasians and African Americans with similar BMIs [29]. Further studies including a large Asian population with a longer follow-up period are needed.

The relation between WC and visceral adipose tissue is known to be influenced by sex as well as age and ethnicity [30]. In our study, the associations between WC and mortality were different by sex. WC was associated with overall mortality with a reverse J-shaped association in men, whereas the association between WC and overall mortality was not significant in women. Consistent with the results of our study, the Melbourne Collaborative Cohort study showed a difference between WC and mortality by sex. In the study, there was a linear association between WC and all-cause mortality for men, whereas a U-shaped association was observed for women among 16,969 men and 24,344 women aged 27 to 75 years [31]. The lack of correlation between overall mortality and WC in women could be due to the fact that WC measurement does not distinguish between visceral fat and subcutaneous fat deposition in the belly. Visceral fat accumulation is widely regarded as a risk factor for cardiovascular diseases and subcutaneous fat is regarded a more benign type of fat.

There were several strengths in our study. This is the first study to estimate the relationship between WC and mortality over a long follow-up period and the association between WC and CVD incidence in a Korean population. In addition, we analyzed the cohort data separately by sex. We excluded subjects who had a past history of cancers, stroke, ischemic heart, or chronic obstructive pulmonary disease or who died within one year from the start of the study because underlying diseases may affect the mortality results. Excluding subjects who died within the first year could have reduced higher risk subjects from the analysis.

Limitations of this study include the fact that we did not classify the causes of death. We did not follow up on changes in WC over the follow-up period. Because the NHIS cohort data were not collected for our study, it was impossible to know the WC at the moment of the event. Although the threshold for risky WC is suggested for men and women separately, the reference WC was set to less than 80 cm for both men and women in our study. Additional analysis setting different reference values of WC for men and women separately is required. Although participants who had chronic diseases such as cancers, stroke, ischemic heart, and chronic obstructive pulmonary disease were excluded from the study, other serious diseases could potentially affect the associations between WC and mortality. Because we did not adjust the socioeconomic state that could affect subject's health, it is possible that the different socioeconomic aspects had acted as confounding variables in our study.

In conclusion, WC exhibited a significant reverse J-shaped association with overall mortality in men, and the risk of CVD incidence showed a positive association with central obesity for both men and women. These findings suggest that measurement of WC in addition to BMI may be needed in consideration of obesity-related health risks, and active interventions to reduce WC would be helpful to prevent CVD. Further research setting different reference values of WC for men and women separately and including changes in WC over the follow-up period is needed.

## Author Contributions

**Conceptualization:** Young Sun Hong, Yeon-Ah Sung, Hyejin Lee.

**Formal analysis:** Do Kyeong Song, Hyejin Lee.

**Investigation:** Do Kyeong Song, Young Sun Hong, Yeon-Ah Sung, Hyejin Lee.

**Methodology:** Young Sun Hong, Yeon-Ah Sung, Hyejin Lee.

**Project administration:** Young Sun Hong, Yeon-Ah Sung, Hyejin Lee.

**Supervision:** Young Sun Hong, Yeon-Ah Sung, Hyejin Lee.

**Validation:** Do Kyeong Song, Young Sun Hong, Yeon-Ah Sung, Hyejin Lee.

**Writing – original draft:** Do Kyeong Song.

**Writing – review & editing:** Do Kyeong Song, Hyejin Lee.

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
