## [Decision Letter · Decision Letter 0]

12 Feb 2022

PONE-D-21-39364Waist circumference and mortality or cardiovascular events in a general Korean populationPLOS ONE

Dear Dr. Lee,

Thank you for submitting your manuscript to PLOS ONE. After careful consideration, we feel that it has merit but does not fully meet PLOS ONE’s publication criteria as it currently stands. Therefore, we invite you to submit a revised version of the manuscript that addresses the points raised during the review process.The revised version should address all comments.

We look forward to receiving your revised manuscript.

Kind regards,

Petri Böckerman

Academic Editor

PLOS ONE

Journal Requirements:

Reviewers' comments:

Reviewer's Responses to Questions

**Comments to the Author**

1. Is the manuscript technically sound, and do the data support the conclusions?

Reviewer #1: Yes

Reviewer #2: Yes

2. Has the statistical analysis been performed appropriately and rigorously? 

Reviewer #1: Yes

Reviewer #2: Yes

3. Have the authors made all data underlying the findings in their manuscript fully available?

Reviewer #1: Yes

Reviewer #2: Yes

4. Is the manuscript presented in an intelligible fashion and written in standard English?

Reviewer #1: Yes

Reviewer #2: Yes

5. Review Comments to the Author

Reviewer #1: In this paper, the authors examined the associations between waist circumference (WC) and mortality or CVD incidence in a subset of Korean National Health Insurance Service cohort. They found that CVD incidence, but not mortality, was positively associated with WC in both men and women.

The paper is interesting for the ample number of subjects studied; nevertheless, it arises several observations that the authors should address:

1) It seems that the groups are not well balanced in participant numerosity, being the lowest WC group 38% of the total sample, whereas the highest WC group represents only 6% of it.

2) In the majority of the international consensus (e.g.: IDF’s), the threshold for risky WC for Asian population is set in 80 cm for women and 90 cm for men. Therefore:

- a) In men, choosing the group with range 85-90 cm would have been more appropriate as a reference, rather than <80 category. Indeed, a low (< 80 cm) WC in men could be due to underlying disease, malnutrition or different socioeconomic aspects that could affect subject’s health.

- b) The J or U shaped curve is a common finding in the relationships in which weight or other anthropometric variables related to nutritional status are considered. The point is to look at the nadir of the curve and consider if it falls within a “physiologic” status or not.

- c) The lack of correlation between overall mortality and WC in women could be due to the fact that WC measurement does not distinguish between visceral fat and subcutaneous fat deposition in the belly (the last one being a more benign type of fat).

3) Perhaps the decision of excluding people who died within the first year could have reduced higher risk subjects from the analysis.

4) It is not known the WC at the moment of the event (death or ACV accident): this data would had given firmer conclusions.

5) Figures:

In fig 1, the horizontal axis is very expanded and the results are presented as cubic spline interpolation, whereas in fig. 2 the data are presented as points. Please explain the reason of such a different presentation.

Reviewer #2: It is an interesting article, but the following points need to be corrected.

1. The authors reported that they have received funding for this study. However, the source of fund was not mentioned in the article

2. Abstract: is preferred to write the full word of acronym when it first appears in the text.

3. Introduction: ok

4. Methods: ok

5. Results: make the tables and figures self-explanatory by including all necessary thing in the title

6. Discussion: OK

7. Conclusion: the conclusion is in line with the result, but what would the author recommended for the future researchers?

6. PLOS authors have the option to publish the peer review history of their article (what does this mean?). If published, this will include your full peer review and any attached files.

Reviewer #1: **Yes: **Raffaele Carraro Casieri

Reviewer #2: No

---

## [Author Response · Author response to Decision Letter 0]

17 Mar 2022

PONE-D-21-39364

Waist circumference and mortality or cardiovascular events in a general Korean population

PLOS ONE

Dear Dr. Lee,

Thank you for submitting your manuscript to PLOS ONE. After careful consideration, we feel that it has merit but does not fully meet PLOS ONE’s publication criteria as it currently stands. Therefore, we invite you to submit a revised version of the manuscript that addresses the points raised during the review process.

The revised version should address all comments.

→ Thank you for your kind advice. We inserted the statement of our financial disclosure as “The authors received no specific funding for this work.” in our cover letter.

We look forward to receiving your revised manuscript.

Kind regards,

Petri Böckerman

Academic Editor

PLOS ONE

Journal Requirements:

→ Thank you for your kind advice. As you recommended, we ensure that our manuscript meets PLOS ONE's style requirements, including those for file naming.

Reviewers' comments:

Reviewer's Responses to Questions

Comments to the Author

1. Is the manuscript technically sound, and do the data support the conclusions?

Reviewer #1: Yes

Reviewer #2: Yes

2. Has the statistical analysis been performed appropriately and rigorously?

Reviewer #1: Yes

Reviewer #2: Yes

3. Have the authors made all data underlying the findings in their manuscript fully available?

Reviewer #1: Yes

Reviewer #2: Yes

4. Is the manuscript presented in an intelligible fashion and written in standard English?

Reviewer #1: Yes

Reviewer #2: Yes

5. Review Comments to the Author

Reviewer #1: In this paper, the authors examined the associations between waist circumference (WC) and mortality or CVD incidence in a subset of Korean National Health Insurance Service cohort. They found that CVD incidence, but not mortality, was positively associated with WC in both men and women.

The paper is interesting for the ample number of subjects studied; nevertheless, it arises several observations that the authors should address:

1) It seems that the groups are not well balanced in participant numerosity, being the lowest WC group 38% of the total sample, whereas the highest WC group represents only 6% of it.

→ Thank you for your kind comment. In this study, quintiles for WC were not obtained. Furthermore, men and women were not distinguished in the analysis of basic characteristics. “Although the threshold for risky WC is suggested for men and women separately, the reference WC was set to less than 80 cm for both men and women in our study. Additional analysis setting different reference values of WC for men and women separately is required.” had been inserted in the eighth paragraph entitled “Discussion”.

2) In the majority of the international consensus (e.g.: IDF’s), the threshold for risky WC for Asian population is set in 80 cm for women and 90 cm for men. Therefore:

- a) In men, choosing the group with range 85-90 cm would have been more appropriate as a reference, rather than <80 category. Indeed, a low (< 80 cm) WC in men could be due to underlying disease, malnutrition or different socioeconomic aspects that could affect subject’s health.

→ Thank you for your kind comment. As you recommended, we fully agree that additional analysis is required by setting different reference values of WC for men and women separately. We excluded subjects who had a past history of cancers, stroke, ischemic heart, or chronic obstructive pulmonary disease or who died within one year from the start of the study because underlying diseases may affect the mortality results. Therefore, we think subjects with underlying disease or malnutrition are unlikely to have been included in our study. However, it is possible that different socioeconomic aspects could affect the results of our study. “Because we did not adjust the socioeconomic state that could affect subject’s health, it is possible that the different socioeconomic aspects had acted as confounding variables in our study.” has been inserted in the eighth paragraph entitled “Discussion”.

- b) The J or U shaped curve is a common finding in the relationships in which weight or other anthropometric variables related to nutritional status are considered. The point is to look at the nadir of the curve and consider if it falls within a “physiologic” status or not.

→ Thank you for your kind comment. To date, there have been several studies about the association between WC and mortality; however, the results have been inconsistent as described in the discussion. The results of our study had shown that WC exhibited a significant reverse J-shaped association with overall mortality in men. We excluded subjects who had a past history of cancers, stroke, ischemic heart, or chronic obstructive pulmonary disease or who died within one year from the start of the study because underlying diseases may affect the mortality results. We think that excluding higher risk subjects from the analysis makes the nadir of the curve about the association between WC and mortality or cardiovascular events to the “physiologic” status in our study. However, additional confounding factors such as the different socioeconomic aspects or other serious diseases could potentially affect the associations between WC and mortality. 

- c) The lack of correlation between overall mortality and WC in women could be due to the fact that WC measurement does not distinguish between visceral fat and subcutaneous fat deposition in the belly (the last one being a more benign type of fat).

→ We appreciate for your advice. As you recommended, “The lack of correlation between overall mortality and WC in women could be due to the fact that WC measurement does not distinguish between visceral fat and subcutaneous fat deposition in the belly. Visceral fat accumulation is widely regarded as a risk factor for cardiovascular diseases and subcutaneous fat is regarded a more benign type of fat.” had been inserted sixth paragraph entitled “Discussion”.

3) Perhaps the decision of excluding people who died within the first year could have reduced higher risk subjects from the analysis.

→ Thank you for your kind comment. “Excluding subjects who died within the first year could have reduced higher risk subjects from the analysis.” has been inserted in the seventh paragraph entitled “Discussion”.

4) It is not known the WC at the moment of the event (death or ACV accident): this data would had given firmer conclusions.

→ Thank you for your kind comment. We used the cohort database released by the NHIS including individuals older than 40 years who were invited to participate in a biannual health checkup. “Because the NHIS cohort data were not collected for our study, it was impossible to know the WC at the moment of the event.” has been inserted in the eighth paragraph entitled “Discussion”. “Further research setting different reference values of WC for men and women separately and including changes in WC over the follow-up period is needed.” has been inserted in the last paragraph entitled “Discussion”.

5) Figures:

In fig 1, the horizontal axis is very expanded and the results are presented as cubic spline interpolation, whereas in fig. 2 the data are presented as points. Please explain the reason of such a different presentation.

→ Thank you for your kind comment. Figure 1 represents the mortality according to WC as a continuous variable, while Figure 2 shows the CVD incidence according to WC as a categorical variable. We had changed the “Association between waist circumference categories and overall mortality and CVD mortality as a cubic spline after adjusting for age” to “Association between waist circumference as a continuous variable and overall mortality and CVD mortality as a cubic spline after adjusting for age” in the title of figure 1. 

Reviewer #2: It is an interesting article, but the following points need to be corrected.

1. The authors reported that they have received funding for this study. However, the source of fund was not mentioned in the article

→ Thank you for your kind comment. We changed the answer of the question about Financial Disclosure from “The funders had no role in study design, data collection and analysis, decision to publish, or preparation of the manuscript.” to “The authors received no specific funding for this work.” 

2. Abstract: is preferred to write the full word of acronym when it first appears in the text.

→ Thank you for your kind comment. As you recommended, we changed WC to waist circumference (WC) when it first appears in the abstract.

3. Introduction: ok

4. Methods: ok

5. Results: make the tables and figures self-explanatory by including all necessary thing in the title

→ Thank you for your kind comment. 

We had changed “Demographic and clinical characteristics of the subjects” to “Demographic and clinical characteristics of the subjects according to waist circumference categories” in the title of table 1. 

Figure 1 represents the mortality according to WC as a continuous variable, while Figure 2 shows the CVD incidence according to WC as a categorical variable. We had changed the “Association between waist circumference categories and overall mortality and CVD mortality as a cubic spline after adjusting for age” to “Association between waist circumference as a continuous variable and overall mortality and CVD mortality as a cubic spline after adjusting for age” in the title of figure 1. 

6. Discussion: OK

7. Conclusion: the conclusion is in line with the result, but what would the author recommended for the future researchers?

→ Thank you for your kind advice. As you recommended, “Further research setting different reference values of WC for men and women separately and including changes in WC over the follow-up period is needed.” has been inserted in the last paragraph entitled “Discussion”.

6. PLOS authors have the option to publish the peer review history of their article (what does this mean?). If published, this will include your full peer review and any attached files.

Do you want your identity to be public for this peer review? For information about this choice, including consent withdrawal, please see our Privacy Policy.

Reviewer #1: Yes: Raffaele Carraro Casieri

Reviewer #2: No

→ Thank you for your kind comment. As you recommended, we uploaded our figure files to the Preflight Analysis and Conversion Engine (PACE) digital diagnostic tool.

---

## [Decision Letter · Decision Letter 1]

12 Apr 2022

Waist circumference and mortality or cardiovascular events in a general Korean population

PONE-D-21-39364R1

Dear Dr. Lee,

We’re pleased to inform you that your manuscript has been judged scientifically suitable for publication and will be formally accepted for publication once it meets all outstanding technical requirements.

Kind regards,

Petri Böckerman

Academic Editor

PLOS ONE

Additional Editor Comments (optional):

Reviewers' comments:

Reviewer's Responses to Questions

**Comments to the Author**

1. If the authors have adequately addressed your comments raised in a previous round of review and you feel that this manuscript is now acceptable for publication, you may indicate that here to bypass the “Comments to the Author” section, enter your conflict of interest statement in the “Confidential to Editor” section, and submit your "Accept" recommendation.

Reviewer #1: All comments have been addressed

Reviewer #2: All comments have been addressed

2. Is the manuscript technically sound, and do the data support the conclusions?

Reviewer #1: Yes

Reviewer #2: (No Response)

3. Has the statistical analysis been performed appropriately and rigorously? 

Reviewer #1: Yes

Reviewer #2: (No Response)

4. Have the authors made all data underlying the findings in their manuscript fully available?

Reviewer #1: Yes

Reviewer #2: (No Response)

5. Is the manuscript presented in an intelligible fashion and written in standard English?

Reviewer #1: Yes

Reviewer #2: (No Response)

6. Review Comments to the Author

Reviewer #1: (No Response)

Reviewer #2: (No Response)

7. PLOS authors have the option to publish the peer review history of their article (what does this mean?). If published, this will include your full peer review and any attached files.

Reviewer #1: **Yes: **Raffaele Carraro Casieri

Reviewer #2: No

---

## [Editor Report · Acceptance letter]

18 Apr 2022

PONE-D-21-39364R1 

Waist circumference and mortality or cardiovascular events in a general Korean population 

Dear Dr. Lee:

I'm pleased to inform you that your manuscript has been deemed suitable for publication in PLOS ONE. Congratulations! Your manuscript is now with our production department. 

Kind regards, 

on behalf of

Professor Petri Böckerman 

Academic Editor

PLOS ONE